# Gut Microbiota as a Potential Predictive Biomarker in Relapsing-Remitting Multiple Sclerosis

**DOI:** 10.3390/genes13050930

**Published:** 2022-05-23

**Authors:** Vicente Navarro-López, María Ángeles Méndez-Miralles, Rosa Vela-Yebra, Ana Fríes-Ramos, Pedro Sánchez-Pellicer, Beatriz Ruzafa-Costas, Eva Núñez-Delegido, Humberto Gómez-Gómez, Sara Chumillas-Lidón, Jose A. Picó-Monllor, Laura Navarro-Moratalla

**Affiliations:** 1Ph.D. Program in Health Sciences, Campus de los Jerónimos 135, UCAM-Universidad Católica San Antonio de Murcia, 30107 Murcia, Spain; 2MiBioPath Research Group, Department of Clinical Medicine, Campus de los Jerónimos 135, UCAM-Universidad Católica San Antonio de Murcia, 30107 Murcia, Spain; pedro.sanchez@bioithas.com (P.S.-P.); beatriz.ruzafa@bioithas.com (B.R.-C.); eva.nunez@bioithas.com (E.N.-D.); jhumbert.gomez@gmail.com (H.G.-G.); sara.chumillas@bioithas.com (S.C.-L.); japico@umh.es (J.A.P.-M.); laura.navarro@bioithas.com (L.N.-M.); 3Infectious Disease Unit, University Hospital Vinalopó, Carrer Tonico Sansano Mora 14, 03293 Elche, Spain; 4Department of Neurology, University Hospital of Torrevieja, Carretera CV95, s/n, 03186 Alicante, Spain; pradipika12@gmail.com; 5Department of Neurology, University Hospital of Vinalopó, Carrer Tonico Sansano Mora 14, 03293 Elche, Spain; anafries@hotmail.com; 6Department of Pharmacology, Pediatrics and Organic Chemistry, Faculty of Pharmacy, Universidad Miguel Hernández de Elche, 03202 Elche, Spain

**Keywords:** gut microbiota, microbiome, active relapsing-remitting multiple sclerosis, *Ezakiella*, *Bilophila*

## Abstract

Background: The influence of the microbiome on neurological diseases has been studied for years. Recent findings have shown a different composition of gut microbiota detected in patients with multiple sclerosis (MS). The role of this dysbiosis is still unknown. Objective: We analyzed the gut microbiota of 15 patients with active relapsing-remitting multiple sclerosis (RRMS), comparing with diet-matched healthy controls. Method: To determine the composition of the gut microbiota, we performed high-throughput sequencing of the 16S ribosomal RNA gene. The specific amplified sequences were in the V3 and V4 regions of the 16S ribosomal RNA gene. Results: The gut microbiota of RRMS patients differed from healthy controls in the levels of the *Lachnospiraceae*, *Ezakiella*, *Ruminococcaceae*, *Hungatella*, *Roseburia*, *Clostridium*, *Shuttleworthia*, *Poephyromonas*, and *Bilophila* genera. All these genera were included in a logistic regression analysis to determine the sensitivity and the specificity of the test. Finally, the ROC (receiver operating characteristic) and AUC with a 95% CI were calculated and best-matched for *Ezakiella* (AUC of 75.0 and CI from 60.6 to 89.4) and *Bilophila* (AUC of 70.2 and CI from 50.1 to 90.4). Conclusions: There is a dysbiosis in the gut microbiota of RRMS patients. An analysis of the components of the microbiota suggests the role of some genera as a predictive factor of RRMS prognosis and diagnosis.

## 1. Introduction

Multiple sclerosis (MS) is a rare chronic autoimmune inflammatory and degenerative disease characterized by a multifocal demyelination of the central nervous system (CNS) [1]. Traditionally, incidence rates have been higher in areas away from the equator and in developed countries [2,3]. This geographical distribution has been related to differences in sun exposure and vitamin D levels, but it could also be related to the hygiene hypothesis that correlates excess sterilization, cleaning, vaccination, and the widespread use of antibiotics to the complication of proper development of immune responses, which could play an important role in the development of MS [4].

The disease typically evolves as relapses that consist of acute episodes of demyelination of the white matter, which manifest as an altered nerve conduction, secondary axonal damage, oligodendrocyte loss, and subsequent permanent damage of the involved CNS area [5]. This alteration is immunomodulated by myelin-autoreactive CD4+ T cells [6], although a recent report suggested that B cells could also be involved in the pathogenesis of this neurological disease [7]. The etiology of MS remains unknown, and it is postulated that it may developed in genetically predisposed individuals due to an alteration of the immune system triggered by an unknown environmental factor not yet identified. Some studies have evaluated the influence of the Epstein–Barr virus, smoking, low levels of vitamin D, or alcohol consumption [8]. Among them, a new one emerges as a potential factor in the etiology of MS: gut microbiota. An increasingly growing number of reports published over the last years have suggested the potential role of microbiota in MS [9].

MS is the most frequent demyelinating disease, which usually develops in young adults between the second and the third decade of life, being more frequent in females than in males [1]. The incidence is growing in western countries, but also the difference between both sexes (proportion of female/male of 2.3–3.5:1). There are some theories to try to explain these differences, but they probably are based on the difference in sexual hormones (progesterone and some estrogens are protectives factors in pregnancy), more susceptibility to vitamin D deficiency, other environmental risk factors, and more genetic predisposition with an overrepresentation of MS susceptibility after the random inactivation of one X chromosome in women [10].

To prevent disease activity and disability progression, new disease-modifying therapies (DMTs) have emerged based on controlling the altered immune response in MS patients. The first therapies that were effective, beta-interferon and glatiramer acetate, had immunomodulatory characteristics, just as teriflunomide did later (an immunosuppressive drug that interfered in pyrimidine synthesis), all with modest efficacy. Afterwards, some DMTs have appeared with different therapeutic immune targets. Dimethyl fumarate has anti-inflammatory and cytoprotective properties with moderate efficacy. Other DMTs work by blocking the access of activated lymphocytes to the CNS with two strategies: restricting their migration across the blood–brain barrier by blocking adhesion α4β1 integrin (natalizumab), or avoiding their egress from lymph nodes by inhibition of the sphingosine-1-phosphate receptor (fingolimod and siponimod). On the other hand, some DMTs (ocrelizumab and ofatumumab) are monoclonal antibodies that bind the CD20 antigen located on the surface of mature B cells and produce their depletion, causing chronic immunosuppression [11]. Lastly, immune reconstitution therapies have appeared which cause a more aggressive state of immunosuppression followed by reconstitution with a new profile of the immune system, including alemtuzumab, a monoclonal antibody binding the CD52 antigen expressed on the surface of T and B lymphocytes, and cladribine, a synthetic chlorinated deoxyadenosine analog that interferes with DNA synthesis and repair, causing the sustained reduction of circulating T and B lymphocytes [12].

Deciding which DMT should be provided and when for each patient is relevant to control disease progression and to improve the patient’s quality of life [13]. To date, there are scarce resources and tools in clinical practice that can be used for MS diagnosis and to predict an adequate response to treatment [14]. Most physicians rely on magnetic resonance imaging (MRI) scans to monitor the brain damage of patients under treatment. However, this approach requires monitoring brain damage for several months before determining treatment failure, a time where neuronal deterioration has already occurred. Additionally, other solutions exist, such as cerebrospinal fluid (CSF) analysis or blood tests, that are suboptimal due to their invasiveness or low specificity in predicting MS treatment response. Taking into consideration the importance of early personalized treatment to stop progression, gut microbiota may be an innovative approach in the identification of novel biomarkers used to predict MS diagnosis and the patient’s response to DMT. Gut microbiota has a population of around 100 trillion microorganisms mainly composed by bacteria, but it also contains archaea, eukaryotes, and viruses [15]. It begins to colonize the gastrointestinal tract at the moment of birth, and the initial composition depends on the mode of delivery, since the starting population comes from the maternal urogenital tract or the skin in vaginal or caesarean section delivery, respectively [16]. The composition of the microbiota is defined after our first years of life but could be changed by different environmental factors [17]. An increasing number of evidence is starting to correlate gut microbiota with several diseases, such as obesity [18] and inflammatory bowel disease [19], among others. In addition, it has been shown that gut microbiota plays a crucial role for the host metabolism and also in the development, maturation, and appropriate function of the immune system, preventing colonization by pathologic microorganisms [20] and helping to maintain the intestinal immune barrier by stimulating innate [21] and adaptive immune responses [22]. Moreover, the fact that MS is related with some gastrointestinal symptoms such as constipation, fecal incontinence, and intestinal permeability suggests a possible connection between gut microbiota and demyelinating diseases [23]. Recently, specific functions have started to be associated with different microorganisms, such as the potential role of *Bacteroides fragilis* to induce Foxp3+ Treg cells. These cells can ameliorate inflammation conditions in ulcerative colitis or experimental autoimmune encephalomyelitis (EAE), a murine model of MS [24]. In fact, several studies of EAE have shown a possible correlation between gut microbiota and MS. These studies have described how the use of broad-spectrum antibiotics is related with a slower progression of EAE and a decrease in proinflammatory cytokines and mesenteric Th17 cells [25], as well as how gut microbiota can influence the permeability of the blood–brain barrier [26]. These data have suggested that gut microbiota can modify a host’s immune response in EAE studies, influencing its origin, prevention, and possible treatment. Due to the increasing number of scientific studies that have identified gut dysbiosis in the stool samples of MS patients, we decided to investigate the possible role of the gut microbiota composition as a predictive factor of the disease in a prospective 24-month follow-up study with a cohort of patients with active relapsing-remitting multiple sclerosis (RRMS).

## 2. Material and Method

### 2.1. Study Population

The patients evaluated for inclusion in this study had previously been diagnosed with RRMS according to the 2017 McDonald criteria from clinical and radiological findings [27]. All the patients included were diagnosed with active RRMS, defined in our protocol as the presence of at least one relapse (defined as a new symptom or worsening of a previous one with objective findings reflecting focal or multifocal involvement of the CNS, developing acutely or subacutely, lasting greater than 24 h without a coexisting fever or infection, and followed by a full or incomplete recovery), at least one T1-gadolinium-enhancing (GdE) lesion, or new or unequivocally enlarging T2 lesions documented by magnetic resonance imaging (MRI) [28] in the two years prior to their inclusion in the study. The control group was selected from the healthy family members or partners who lived with the patients to homogenize both groups, especially matched by diet. All the participants provided sociodemographic and other clinical data, which included age, sex, body mass index (BMI), and the usual diet they followed during the last year. The exclusion criteria were the following: use of immunosuppressant drugs such as systemic corticosteroids, methotrexate, cyclosporine, or anti-tumor necrosis factor alpha (TNFα) prescribed for any comorbidity in the previous 3 months; systemic antibiotics or demonstration of a bacterial infection in the previous 2 weeks; and concomitant diagnosis of cirrhosis or intestinal bowel disease.

Furthermore, the MS patient group continued to be evaluated periodically for a 24-month follow-up period by an expert neurologist in MS. Moreover, brain MRI (and spinal cord MRI in some cases) was performed at least every 12 months to detect the presence of new lesions in the follow-up period. Expert neuroradiologists manually counted the number of new or unequivocally enlarging T2 lesions and T1-GdE lesions detected by MRI. Stool samples were obtained from all the participants and were immediately frozen and stored at −80 °C until they were processed for high-throughput sequencing. The Ethics Committees at the University Hospital of Vinalopó in Elche and the University Hospital of Torrevieja, Spain, approved the study in accordance with the Helsinki Declaration protocol, and all the subjects gave their written, informed consent before being included in the study.

### 2.2. High-Throughput Sequencing of Stool Samples of Patients with MS

DNA from the stool samples was isolated with the aid of a MagnaPure Compact System (Roche Life Science, Mannheim, Germany) to avoid bias in the DNA purification toward the misrepresentation of Gram-positive bacteria, following the Yuan et al., method [29]. For high-throughput sequencing, the hypervariable region of V3-V4 of the bacterial 16s ribosomal RNA gene was amplified using key-tagged eubacterial primers [30] and sequenced with a MiSeq Illumina Platform (Illumina, San Diego, CA, USA). The Illumina recommendations for library preparation and sequencing for metagenomics studies were used. The bacterial compositions of all the patients were compared with a group of healthy diet-matched individuals selected from the healthy partners living with them.

### 2.3. Bioinformatics

The resulting sequences were split considering the barcode introduced during the PCR reaction, while the R1 and R2 reads were overlapped using PEAR program version 0.9.122 (https://cme.h-its.org/exelixis/web/software/pear/doc.html, accessed on 4 September 2019), providing a single FASTQ file for each sample. Quality control of the sequences was performed using different steps. First, quality filtering (with a minimum threshold of Q20) was performed using the fastx tool kit version 0.013 (http://hannonlab.cshl.edu/fastx_toolkit/download.html, accessed on 4 September 2019). Finally, primer (16s rRNA primers) trimming and length selection (reads over 300nts) was performed with cutadapt version 1.4.123 (https://cutadapt.readthedocs.io/en/stable/, accessed on 4 September 2019). These FASTQ files were converted to FASTA files, and UCHIME program version 7.0.1001 (https://www.drive5.com/usearch/manual/uchime_algo.html, accessed on 4 September 2019) was used in order to remove chimeras that could arise during the amplification and sequencing step. These clean FASTA files were compared with BLAST24 against the NCBI 16s rRNA database using blast version 2.2.29+ (https://blast.ncbi.nlm.nih.gov/Blast.cgi, accessed on 4 September 2019). The resulting XML files were processed using a python script from the Statistical Department of the Biotech Bioithas (Parc Científic de la Universitat d’Alacant, Alicante, Spain) in order to annotate each sequence at different phylogenetic levels (phylum, family, genera, and species).

### 2.4. Statistical Analysis

The descriptive quantitative variables were expressed as means ± SD (standard deviation) and as frequencies for qualitative variables. Comparisons between the groups were performed with a U Mann–Whitney test for quantitative data, while qualitative variables were analyzed using chi-squared and Fisher tests. An odds ratio (OR) with a 95% confidence interval (CI) was taken as a measure of effect size. A two-tailed p value of less than 0.05 was considered to indicate statistical significance. Finally, a logistic regression analysis model and the area under the receiver operating characteristic curve (AUCROC) using the variable case as dependent and the variable number of sequences of each genera as independent was performed. The predicted values were obtained, and the area under the ROC curve was estimated, as well as the cut-off point that maximized sensitivity and specificity. All the statistical analyses were performed using IBM SPSS statistics version 22 (SPSS Inc. Chicago, IL, USA) and R version 3.2.3 (https://cran.r-project.org/index.html, accessed on 14 October 2019). In the case of microbiota analysis, alpha diversity was conducted using a specaccum program in the vegan package, as implemented for R version 3.2.3.

## 3. Results

From May to June 2016, fifteen patients with a diagnosis of RRMS were enrolled in the study. Thirteen patients (86.6%) were female, with a mean age of 38.15 ± 8.08 years old. All the patients were under disease-modifying therapies (DMTs) when they were included in the study. However, 3/15 patients (20%) had stopped their DMTs weeks or months before they collected the stool samples because they were in a switching period until a new selected treatment started due to uncontrolled disease activity. Beta-interferon (6/15; 40.0%) and fingolimod (4/15; 26.7%) were the most frequent treatments administered. All the patients met the 2017 McDonald criteria of RRMS [27] at the time of inclusion in the study, myelitis being the most prevalent type of initial clinical presentation (6/15; 40.0%). The baseline characteristics of the RRMS patients are described in Table 1.

The compositions of the gut microbiota in all the patients were analyzed by high-throughput sequencing. This allowed us to describe “relapsing-remitting multiple sclerosis microbiota” based on the median values of the bacteria detected in all the patients (Appendix A). The analysis of the levels of species showed 112 different bacteria in the gut microbiota of the RRMS patient group. The most frequent genera detected among the analyzed stool samples were *Bacteroides, Faecalibacterium*, and *Ruminococcus*. Other common detected genera with mean values higher than 1% of the total stool samples were *Gemmiger, Roseburia, Alistipes, Eubacterium, Oscillibacter, Lachnoclostridium, Barnesiella, Parabacteroides, Ruminiclostridium, Blautia, Phascolarctobacterium, Erysipelatoclostridium*, and *Bifidobacterium*.

The bacterial compositions of all the RRMS patients were compared with a control group composed of healthy partners living with the patients to homogenize both groups by age and diet. The Shannon biodiversity index was calculated and then compared (mean, (interquartile range—IQR)) between the RRMS patients (3.05, (2.86–3.31)) and the healthy controls (2.43, (2.19–2.57)), showing significantly higher variability in the control group (*p* < 0.01) (Figure 1).

To search for possible genera that could allow the clustering of patient data, we performed a principal component analysis of each patient, comparing bacteria composition at the taxonomical genera level. No significant differences were found at the microbial structure level between both groups (Figure 2).

In addition, a logistic regression analysis was performed at the genera level of the microbiota composition and detected statistically significant differences between these two population groups with a *p* value < 0.05 in levels of the genera of *Lachnospiraceae, Ezakiella, Ruminococcaceae, Hungatella, Roseburia, Clostridium, Shuttleworthia, Porphyromonas*, and *Bilophila* (Table 2). The sensitivity and the specificity of the test, selecting the best cut-off point for the number of the sequences for each germ in the ROC curves (receiver operating characteristic) and analysis of the area under the curve (AUC) with a 95% CI, were calculated for all the above described genera and best-matched for *Ezakiella* (AUC 75.0; CI 60.6 to 89.4) and *Bilophila* (AUC 70.2; CI 50.1 to 90.4) (Figure 3 and Figure 4).

The *Ezakiella* genus was detected in eight (57.1%) cases and in one (7.1%) control. The median sequence numbers between the cases and controls were three and zero, respectively (*p* = 0.004). Regarding the *Bilophila* genus, sequences were found in 13 (92.9%) cases and in 11 (78.6%) controls, the median sequence numbers in cases and controls being 96 and 24, respectively (*p* = 0.036) (Appendix A).

The different components of diet were registered for each patient and control included in the study. No difference was found in the comparative analysis of the main components of the diet in both study groups (Figure 5).

## 4. Discussion

During the last years, several studies have evaluated gut microbial composition in different cohorts of patients with MS. The results have suggested a different microbiota composition in MS patients compared to healthy populations [31,32,33,34,35]. In our study, this analysis showed an important dysbiosis in the group of patients with relapsing-remitting multiple sclerosis (RRMS) compared to healthy controls.

The most important contribution of our work was the study of the gut microbiota composition as a predictive factor of the disease course in patients with active RRMS. We found a statistically significant positive correlation between the number of sequences in the RRMS group and the levels of the *Ezakiella* and *Bilophila* genera. In the ROC curve, we found a cut-off point that established a sensitivity and specificity relation that better discriminated between cases of RRMS and healthy people.

Horton et al. [31], found an unspecified member of *Coriobacteriales* denominated *SV_520*, which was associated with an increased risk of experiencing disease activity (both clinical relapses and new lesions on MRI) in pediatric MS patients. On the other hand, their study showed some bacterial genera as protective factors, such as *Butyricicoccus desmolans*, *Odoribacter splanchnicus*, unidentified species in the *Lachnospiraceae NK4A136 group*, and *Ruminococcaceae*. Another study by Tremlett et al. [32] linked the absence of *Fusobacteria* with a shorter time of relapse, also in pediatric MS cases.

Among the 15 patients described here, we found decreased levels of *Prevotella*, as described previously in other studies with MS patients [33]. We also found low levels of *Streptococcus* genera, data previously reported in several studies that showed low butyrate production in the gut microbiomes of patients with inflammatory diseases [34]. Also, this data may be of interest considering the recent publication of Tankou et al. [35]. These authors published an interventional study evaluating the biological effect of the administration of an oral probiotic mix containing *Streptococcus*, among others, after the completion of 2 months of treatment. Although the number of MS patients included seems to be small, the investigators demonstrated an increase in the relative abundance of species in the gut microbiota and the induction of an anti-inflammatory peripheral immune response with a decrease in proinflammatory monocytes, as well as in activated monocyte and dendritic cell markers, present both in MS patients and healthy controls after the administration of the probiotic mix.

Cantarel et al., found that 15 patients with MS had decreased basal levels of the *Bacteroidaceae* family and the *Faecalibacterium* genera and increased levels of the *Ruminococcus* genera compared with a group of healthy subjects. After supplementation with vitamin D for 90 days, there were significant increased levels of the *Faecalibacterium* genera and *Enterobacteriaceae* family and decreased levels of the *Ruminococcus genera* when they were compared with healthy controls [36].

A larger study compared the gut microbiota of 60 MS patients with 43 healthy controls [37] and evidenced significant higher levels in archaea, especially *Methanobrevibacter smithii* and in the *Akkermansia* genera, among the group of MS patients. These bacteria have been associated with a potential role in inflammation. As in the Cantarel et al., study, the levels of phylum bacteroidetes were reduced in our MS patients, specifically a decrease in the butyrate producer, *Butyricimonas*. Due to the anti-inflammatory properties of butyrate, a reduction in the levels of these species of MS patients may contribute to disease, as occurs in other diseases where microbiota and intestinal permeability seem to be involved [38].

In another study, Tremlett et al. examined the gut microbiota of 18 MS pediatric patients [39] and found increased levels of *Clostridium*, *Bilophila*, *Escherichia*, and *Shigella*. Some strains of these bacteria have been associated with infection and inflammation [40,41]. Tremlett et al. also described a decrease in the levels of *Eubacterium rectale* and *Corynebacterium*. All these observational studies have concluded that MS patients have less biodiversity in the gut microbiota than healthy people and show significant variability in the levels of different bacteria genera, suggesting that may play a role in the inflammatory processes involved in the pathogenesis of the disease.

Finally, we must take some considerations with these results. Significant associations were found with the cross-reactivity of human GDP-L-fucose synthase peptides and immunodominant myelin basic protein peptides by CNS-infiltrating CD4+ T cells implicated in the myelin degradation process typical in MS patients [42]. These autoreactive CD4+T cells can also be activated by homologous GDP-L-fucose synthase from the gut microbiota, especially bacterial species related with MS pathogenesis in previous research [43]. These results coupled with the analysis here communicated and the possible cross-recognition of the GDP-L-fucose synthase enzyme from gut microbiota suggest a possible role of some components of this microbiota as a trigger of pathogenic autoimmune responses in MS. There is scientific evidence indicating that only a few gut bacteria activate this peptide [44]. All this information supports our hypothesis on the role of *Bilophila* and *Ezakiella* levels with disease activity in MS patients.

Another known factor that can affect and modify microbiota is diet. In fact, diet and age are the main factors that contribute to the composition of the microbiota. Although experimental information suggests that there are a variety of dietary interventions that could benefit patients with MS (vitamin D supplementation, low salt intake, etc.) [45], to date, its benefit has not been well-established in either relapsing or progressive MS. Data about the diet and its mean components were collected in both the patient and control groups in our study. Although the control group was made up mostly of the patients’ partners, having chosen nonconsanguineous controls who shared the same diet (no significant differences in the diets of both groups) brought us reliable data obtained through this control group (mainly male). Despite the fact that the influence of gender on the microbiota is important, the most determinant factor of microbiota composition is the diet [46].

So far, to our knowledge, there is no previous work that has studied the microbiota of patients with RRMS by comparing data with a control group matched by diet to avoid the bias of food intake on the analysis of gut microbiota. The results here communicated suggested a relation between higher levels of *Ezakiella* and *Bilophila* genera and RRMS when compared with data for healthy people. The positive correlation here described among these bacterial species of the gut microbiota and the relapses experienced by these patients despite adequate treatment may be relevant and could serve as a predictive biomarker of active RRMS and also a prognosis biomarker in RRMS patients. The main limitations of these results were the low number of analyzed cases and the greater percentage of male patients in the control group. These potential biases should be controlled in future clinical trials that include a greater number of patients and, among these, a greater presence of males to corroborate the data here presented.

## 5. Conclusions

Our study described the gut microbial composition of a case series of active relapsing-remitting MS patients and a control group. Diet data was registered in both groups, and the different components of diet and other related variables such as BMI were similar at baseline. The results suggested that high levels of *Ezakiella* and *Bilophila* genera could be a risk factor in patients with relapsing-remitting MS. The knowledge here communicated holds conceivable promise for developing novel prognosis biomarkers and more efficient therapeutic routes, including the modulation of gut microbiota in RRMS patients.

## Figures and Tables

**Figure 1 genes-13-00930-f001:**
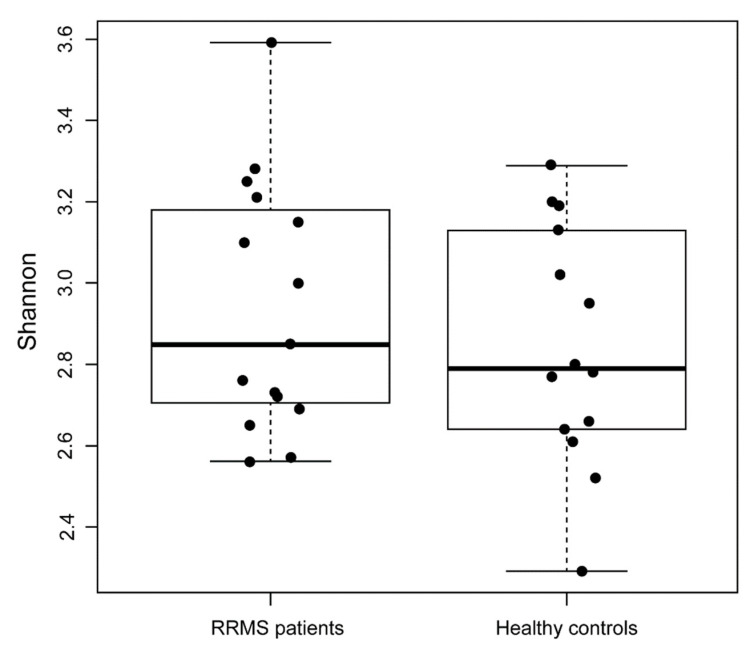
Boxplot based on Shannon diversity index: a comparison between patients with RRMS and healthy controls.

**Figure 2 genes-13-00930-f002:**
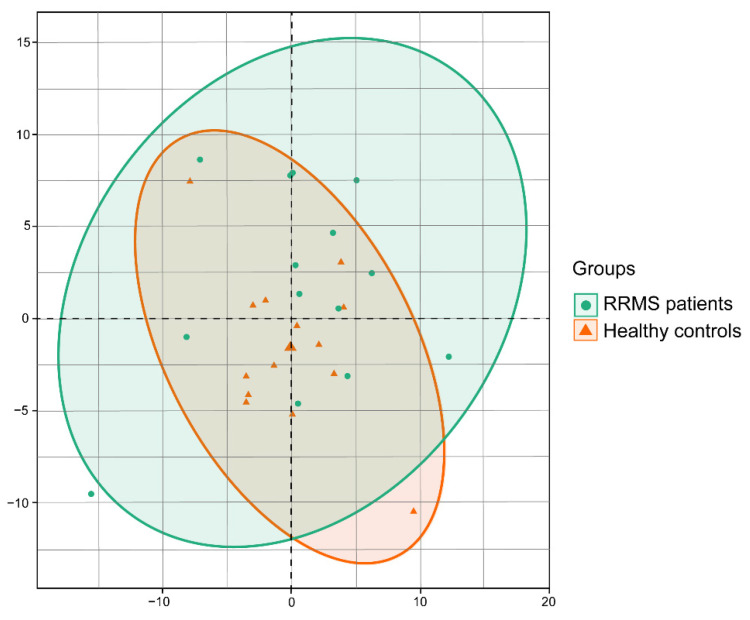
Principal component analysis of the microbiota composition comparing healthy controls and RRMS patients. Comparison of group samples was based on the variability of the bacteria composition at the genera taxonomical level for each RRMS patient and healthy controls.

**Figure 3 genes-13-00930-f003:**
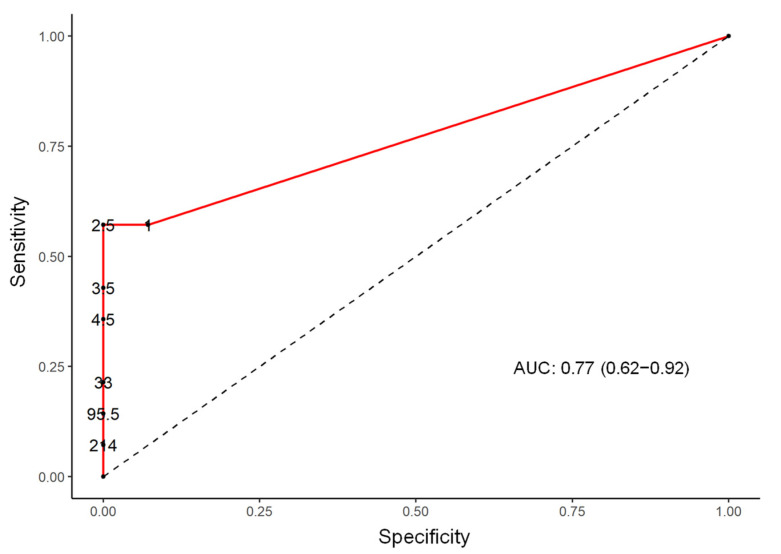
ROC curve (receiver operating characteristic), AUC (area under the curve), and the selection of the best cut-off point with sensitivity and specificity in *Ezakiella* genera.

**Figure 4 genes-13-00930-f004:**
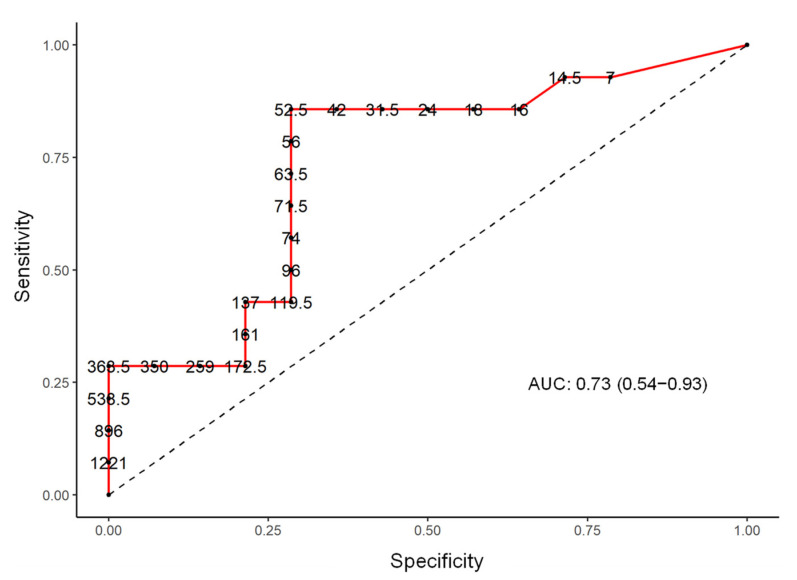
ROC curve (receiver operating characteristic), AUC (area under the curve), and the selection of the best cut-off point with sensitivity and specificity in *Bilophila* genus.

**Figure 5 genes-13-00930-f005:**
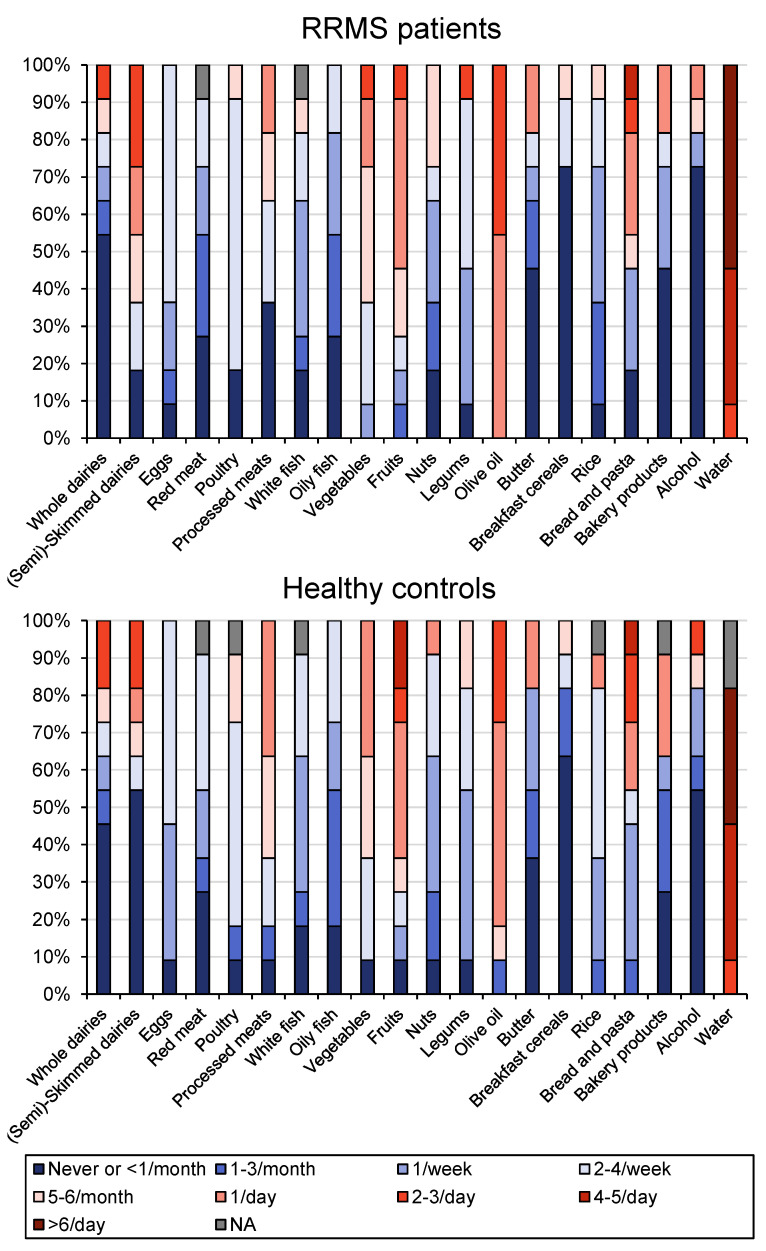
Dietary registry of RRMS patients and healthy controls. Percentage of the main dietary components recorded in the group of patients with active RRMS and healthy controls included in the study.

**Table 1 genes-13-00930-t001:** Demographics of RRMS patients and initial clinical presentation and treatment at baseline.

Patient	Gender	Age (Years)	Weight (kg)	Height (m)	BMI (kg/m^2^)	DMT	Initial Clinical Presentation	EDSS	Relapses	MRI Lesions
1	Female	38	90	1.72	30.42	beta-interferon	Brainstem	2.5	0	0
2	Female	18	57	1.71	19.49	beta-interferon	Myelitis	1	1	1
3	Female	38	57	1.54	22.77	natalizumab	Myelitis	2.5	0	0
4	Female	35	59	1.51	25.88	fingolimod	Myelitis	1	0	0
5	Female	35	107	1.78	33.77	dimethyl fumarate	Myelitis, brainstem	3.5	1	0
6	Female	34	75,5	1.60	29.49	beta-interferon	Optic neuritis	3	3	6
7	Female	31	100	1.80	30.86	fingolimod	Hemiparesis	3.5	2	5
8	Male	33	85.5	1.70	29.58	natalizumab	Myelitis	6.5	0	0
9	Female	45	59	1.65	21.67	fingolimod	Myelitis	2.5	2	8
10	Female	53	92.7	1.55	38.58	beta-interferon	Brainstem	3.5	0	2
11	Female	37	63	1.69	22.05	beta-interferon	Brainstem	1.5	0	0
12	Male	44	92	1.73	30.77	teriflunomide	Hemiparesis	3. 5	2	3
13	Female	29	54	1.60	21.09	dimethyl fumarate	Hemiparesis	1	0	3
14	Female	51	63	1.66	22.86	beta-interferon	Brainstem, optic neuritis	2.5	1	4
15	Female	37	59	1.51	25.88	glatiramer acetate	Brainstem	2	0	0

RRMS: relapsing-remitting multiple sclerosis; BMI: body mass index; DMT: disease-modifying therapy; EDSS: expanded disability status scale; MRI: magnetic resonance imaging.

**Table 2 genes-13-00930-t002:** Case/control ratio by genera with statistical significance difference.

Genera	Multiplier 95% CI	*p*-Value	AUC (95% CI)
*Lachnospiraceae*	0.06 (0.01 to 0.35)	0.001	67.1 (46.7 to 87.6)
*Ezakiella*	13.08 (2.16 to 79.24)	0.005	77.0 (62.0 to 92.0)
*Ruminococcaceae*	12.65 (2.09 to 76.62)	0.006	68.1 (47.9 to 88.3)
*Hungatella*	8.57 (1.41 to 51.89)	0.02	68.3 (51.3 to 85.4)
*Roseburia*	0.12 (0.02 to 0.73)	0.02	68.1 (50.2 to 86.0)
*Clostridium*	7.26 (1.20 to 43.99)	0.03	67.4 (47.5 to 87.2)
*Shuttleworthia*	6.32 (1.04 to 38.29)	0.04	62.4 (42.5 to 82.3)
*Ruminococcaceae*	6.23 (1.03 to 37.71)	0.04	62.4 (42.1 to 82.7)
*Porphyromonas*	6.00 (0.99 to 36.33)	0.05	64.3 (46.0 to 82.5)
*Bilophila*	5.93 (0.98 to 35.91)	0.05	73.0 (54.0 to 93.0)

## Data Availability

The data presented in this study are available on request from the corresponding author. The data are not publicly available due to privacy restrictions.

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
