# Peer review of "Gut Microbiota as a Potential Predictive Biomarker in Relapsing-Remitting Multiple Sclerosis"

_genes, 2022, doi:10.3390/genes13050930_

Round 1
Reviewer 1 Report
This clinical study screened patients with RRMS for microbiota changes. It is interesting to understand the importance of the microbiome for neurological diseases, even if it is difficult to say whether it is a cause or a consequence yet. This study is interesting but can be further improved before publication.
Major concerns
- Please state ethics or patient informed consent.
- The authors stated that certain microbiota genera may be potential biomarkers for predicting prognosis and diagnosis. While no introduction about the present diagnosis methods and their shortness in the manuscript. The readers have to ask why we need the new biomarker. We know, diagnosis via microbiota(sequencing) is not an easy and cheap way. Thus, I think it is necessary to highlight the advantage and importance of new biomarker in the introduction or discussion.
- Most patients are females as they have higher risk. If the healthy controls are from family members or partners, they may be the exact opposite of the patient's gender. Is that a problem?
- Table 2 : Title was missing.
Minor concerns
- line 222, Please use figure 3A, 3B while not figure 3&4. If you still want to use figure 3&4. Please separate them as two individual figures.
- Figure resolution could be further improved.
Author Response
Dear reviewer, thank you very much for your input. Next, we will comment on your comments and suggestions point by point:
Major concerns
Please state ethics or patient informed consent.
This information is included now in the Institutional Review Board Statement : “The study was conducted according to the guidelines of the Declaration of Helsinki and approved by the Ethics Committee for Clinical Research of the University Hospital of Vinalopó and University Hospital of Torrevieja on November 9, 2017.” Information about the written informed consent appears in the first paragraph in the page 5: "... and all subjects gave their written informed consent before being included in the study."
The authors stated that certain microbiota genera may be potential biomarkers for predicting prognosis and diagnosis. While no introduction about the present diagnosis methods and their shortness in the manuscript. The readers have to ask why we need the new biomarker. We know, diagnosis via microbiota(sequencing) is not an easy and cheap way. Thus, I think it is necessary to highlight the advantage and importance of new biomarker in the introduction or discussion.
Definitely it is fundamental give this information to lectures. Is in the fifth paragraph of the introduction. MS diagnosis is based by clinical features and mainly by MRI. The updated diagnostic protocols (McDonald 2017 criteria) provide us with a tool to diagnose as soon as possible MS and be able to apply disease modifying therapies (DMT). Although it is not enough. In the last years there were an effort to find new biomarkers of diagnosis, prognosis or treatment response. Because MS is an autoimmune and also a degenerative disease of central nervous system causing an accumulative disability along the time despite current treatments, we need more prognosis biomarkers to offer the patient the most individualized treatment possible. In our case, recent publications provide us the possible role of the microbiota in the etiology or early stages of the pathogenesis, so that is a great opportunity to find if it can be a early diagnosis and prognosis biomarker test in patients with RRMS.
Most patients are females as they have higher risk. If the healthy controls are from family members or partners, they may be the exact opposite of the patient's gender. Is that a problem?
In the design of our study, we have prioritized the homogeneity of the diet between cases and controls (diet-matched) doing the selection of control group between the partner’s patients in most cases, more than the gender. As we explain in the nineth paragraph of the discussion, we know that there is an influence of the gender in the microbiota, but equal or more important is the diet. We know that it is a limitation of our study, and also the small sample size, so we added this information on discussion part.
Table 2 : Title was missing.
We already included the tittle of table 2.
Minor concerns
line 222, Please use figure 3A, 3B while not figure 3&4. If you still want to use figure 3&4. Please separate them as two individual figures.
We change these figures and now they were separated in the final manuscript.
Figure resolution could be further improved.
The image quality of the figures has been increased to meet the demands of the journal.
Finally, apart from all your suggestions, we made another review of the article in order to improve the quality of scientific contents including table and figures. Moreover, we made the redaction review with a native english translator. We hope this new presentation of our results exceed your quality standards that made it for publication.
Reviewer 2 Report
This manuscript fitted well in the scope of the journal's recent issue on When Genes Meet Microbial Ecology and Evolution. The authors nicely addressed the issue of multiples sclerosis in the manuscript through retrospective analyses of patients and shed the light on the use of gut microflora as indicator of RRMS.
However, sample size (n=15) is too small to make a strong conclusion. The presence of Bilophila/Ezakiella is improbable as a RRMS disease biomarkers considering small sample size. There are some limitations, and corrections are required in the manuscript before its publication.
Considering the rareness of occurrence of MS, in my opinion, the manuscript could be considered for the publication provided that some improvements in the text are required.
Major comments
- Please add few lines in introduction part, regarding the current treatment options and FDA approved therapy for RRMS/MS. Also talk about the immune modulating therapy. Please mentioned the reason for more incidences of MS in female compared to Male, talk about the differences in hormones or genes or immune system differences in both the groups, if any.
- The major limitation of this study is the number of subjects included for MS (n=15). Further, in discussion part the presence of Bilophila/Ezakiella not seems to be reported in any other studies which has been discussed in this manuscript.
Minor comments
Line 40: Please mentioned rare autoimmune disease
Line 213: Please change font to Italic for “Ezakiella and Biophila” here and elsewhere.
Line 214: Please mentioned the Table title is missing
Line 214: Please change Genere to Genera
Line 248: Please change et al to “et al.,” here and elsewhere.
Line 249: Typo change to “double”
Line 251: Change to “genera”
Line 253: Please change relationated to “related” or use other word.
Author Response
Dear reviewer, Thank you very much for your input. Next, we will comment on your comments and suggestions point by point.
Major comments
Please add few lines in introduction part, regarding the current treatment options and FDA approved therapy for RRMS/MS. Also talk about the immune modulating therapy. Please mentioned the reason for more incidences of MS in female compared to Male, talk about the differences in hormones or genes or immune system differences in both the groups, if any.
We tried to improve the introduction part with all these important information. Now there is a good explanation in the fourth paragraph of introduction of all disease modifying therapies which have approved FDA and EMA at the moment and their specific mechanism of action that has an effect on the immune system. Regarding the gender differences in the incidence, we add a brief description in the third paragraph of the novel theories about this event (influence of the gender in the susceptibility of environmental or genetic risk factors in MS)
The major limitation of this study is the number of subjects included for MS (n=15). Further, in discussion part the presence of Bilophila/Ezakiella not seems to be reported in any other studies which has been discussed in this manuscript.
Authors did not find others publications that find relation between Bilophila or Ezakiella and clinical features of the disease. Also, only a few studies found any relation between some genera of the gut microbiota and the activity disease, as we did in our study. This information is in the third paragraph of discussion.
Minor comments
Line 40: Please mentioned rare autoimmune disease
Done, included in the first line of the introduction.
Line 213: Please change font to Italic for “Ezakiella and Biophila” here and elsewhere.
Done, we changed them with the correct nomenclature.
Line 214: Please mentioned the Table title is missing
Done, we added the title.
Line 214: Please change Genere to Genera
Done in the new version of the manuscript.
Line 248: Please change et al to “et al.,” here and elsewhere.
Done in the new version of the manuscript.
Line 249: Typo change to “double”
Done, we change “double” to "increased" in that sentence.
Line 251: Change to “genera”
Done in the new version of the manuscript.
Line 253: Please change relationated to “related” or use other word.
Done. in the new version of the manuscript
Finally, apart from all your suggestions, we made another review of the article in order to improve the quality of scientific contents including table and figures. Moreover, we made the redaction review with a native english translator. We hope this new presentation of our results exceed your quality standards that made it for publication.
Round 2
Reviewer 1 Report
The authors have addressed my concerns.
Reviewer 2 Report
Many questions are answered by the authors, which was raised in the round one of review.
I am satisfied with the changes made by them.